# Bayesian Policy Optimization for Model Uncertainty

**Gilwoo Lee, Brian Hou, Aditya Mandalika Vamsikrishna, Jeongseok Lee, Sanjiban Choudhury, Siddhartha S. Srinivasa**
Paul G. Allen School of Computer Science & Engineering
University of Washington
{gilwoo,bhou,adityavk,jslee02,sanjibac,siddh}@cs.uw.edu

## Abstract

Addressing uncertainty is critical for autonomous systems to robustly adapt to the real world. We formulate the problem of model uncertainty as a continuous Bayes-Adaptive Markov Decision Process (BAMDP), where an agent maintains a posterior distribution over latent model parameters given a history of observations and maximizes its expected long-term reward with respect to this belief distribution. Our algorithm, Bayesian Policy Optimization, builds on recent policy optimization algorithms to learn a universal policy that navigates the exploration-exploitation trade-off to maximize the Bayesian value function. To address challenges from discretizing the continuous latent parameter space, we propose a new policy network architecture that encodes the belief distribution independently from the observable state. Our method significantly outperforms algorithms that address model uncertainty without explicitly reasoning about belief distributions and is competitive with state-of-the-art Partially Observable Markov Decision Process solvers.

## 1 Introduction

At its core, real-world robotics focuses on operating under uncertainty. An autonomous car must drive alongside unpredictable human drivers under road conditions that change from day to day. An assistive home robot must simultaneously infer users' intended goals as it helps them. A robot arm must recognize and manipulate varied objects. These examples share common themes: (1) an underlying dynamical system with unknown *latent parameters* (road conditions, human goals, object identities), (2) an agent that can probe the system via *exploration*, while ultimately (3) maximizing an expected long-term reward via *exploitation*.

The Bayes-Adaptive Markov Decision Process (BAMDP) framework (Ghavamzadeh et al., 2015) elegantly captures the exploration-exploitation dilemma that the agent faces. Here, the agent maintains a *belief*, which is a posterior distribution over the latent parameters $\phi$ given a history of observations. A BAMDP can be cast as a Partially Observable Markov Decision Process (POMDP) (Duff & Barto, 2002) whose state is $(s, \phi)$, where $s$ corresponds to the observable world state. By planning in the belief space of this POMDP, the agent balances explorative and exploitative actions. In this paper, we focus on BAMDP problems in which the latent parameter space is either a *discrete finite set* or a *bounded continuous set* that can be approximated via *discretization*. For this class of BAMDPs, the belief is a categorical distribution, allowing us to represent it using a vector of weights.

The core problem for BAMDPs with continuous state-action spaces is how to explore the reachable belief space. In particular, discretizing the latent space can result in an arbitrarily large belief vector, which causes the belief space to grow exponentially. Approximating the value function over the reachable belief space can be challenging: although point-based value approximations (Kurniawati et al., 2008; Pineau et al., 2003) have been largely successful for approximating value functions of discrete POMDP problems, these approaches do not easily extend to continuous state-action spaces. Monte-Carlo Tree Search approaches (Silver &

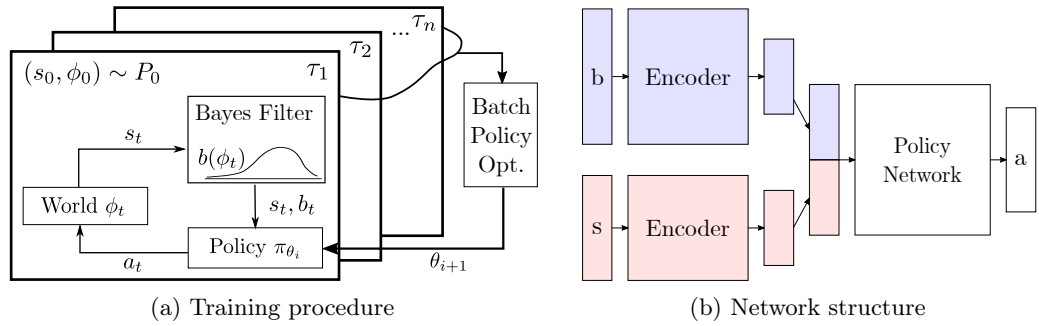

(a) Training procedure  (b) Network structure

Figure 1: An overview of Bayesian Policy Optimization. The policy is simulated on multiple latent models. At each timestep of the simulation, a black-box Bayes filter updates the posterior belief and inputs the state-belief to the policy (Figure 1a). Belief ($b$) and state ($s$) are independently encoded before being pushed into the policy network (Figure 1b)

Veness, 2010; Guez et al., 2012) are also prohibitively expensive in continuous state-action spaces: the width of the search tree after a single iteration is too large, preventing an adequate search depth from being reached.

Our key insight is that we can bypass learning the value function and directly learn a policy that maps beliefs to actions by leveraging the latest advancements in batch policy optimization algorithms (Schulman et al., 2015; 2017). Inspired by previous approaches that train learning algorithms with an ensemble of models (Rajeswaran et al., 2017; Yu et al., 2017), we examine model uncertainty through a BAMDP lens. Although our approach provides only locally optimal policies, we believe that it offers a practical and scalable solution for continuous BAMDPs.

Our method, Bayesian Policy Optimization (BPO), is a batch policy optimization method which utilizes a black-box Bayesian filter and augmented state-belief representation. During offline training, BPO simulates the policy on multiple latent models sampled from the source distribution (Figure 1a). At each simulation timestep, it computes the posterior belief using a Bayes filter and inputs the state-belief pair $(s, b)$ to the policy. Our algorithm only needs to update the posterior along the simulated trajectory in each sampled model, rather than branching at each possible action and observation as in MCTS-based approaches.

Our key contribution is the following. We introduce a Bayesian policy optimization algorithm to learn policies that directly reason about model uncertainty while maximizing the expected long-term reward (Section 4). To address the challenge of large belief representations, we introduce two encoder networks that balance the size of belief and state embeddings in the policy network (Figure 1b). In addition, we show that our method, while designed for BAMDPs, can be applied to continuous POMDPs when a compact belief representation is available (Section 4.2). Through experiments on classical POMDP problems and BAMDP variants of OpenAI Gym benchmarks, we show that BPO significantly outperforms algorithms that address model uncertainty without explicitly reasoning about beliefs and is competitive with state-of-the-art POMDP algorithms (Section 5).

## 2 Preliminaries: Bayesian Reinforcement Learning

The Bayes-Adaptive Markov Decision Process framework (Duff & Barto, 2002; Ross et al., 2008; Kolter & Ng, 2009) was originally proposed to address uncertainty in the transition function of an MDP. The uncertainty is captured by a latent variable, $\phi \in \Phi$, which is either directly the transition function, e.g. $\phi_{sas'} = T(s, a, s')$, or is a parameter of the transition, e.g. physical properties of the system. The latent variable is either fixed or has a known transition function. We extend the previous formulation of $\phi$ to address uncertainty in the reward function as well.

Formally, a BAMDP is defined by a tuple $\langle S, \Phi, A, T, R, P_0, \gamma \rangle$, where $S$ is the observable state space of the underlying MDP, $\Phi$ is the latent space, and $A$ is the action space. $T$ and $R$ are the parameterized transition and reward functions, respectively. The transition function is defined as: $T(s, \phi, a', s', \phi') = P(s', \phi'|s, \phi, a') = P(s'|s, \phi, a')P(\phi'|s, \phi, a', s')$. The initial distribution over $(s, \phi)$ is given by $P_0 : S \times \Phi \to \mathbb{R}^+$, and $\gamma$ is the discount.

Bayesian Reinforcement Learning (BRL) considers the long-term expected reward with respect to the uncertainty over $\phi$ rather than the true (unknown) value of $\phi$. The uncertainty is represented as a *belief distribution* $b \in B$ over latent variables $\phi$. BRL maximizes the following Bayesian value function, which is the expected value *given the uncertainty*:

$$
\begin{aligned}
V_\pi(s, b) &= R(s, b, a') + \gamma \sum_{s' \in S, b' \in B} P(s', b'|s, b, a')V_\pi(s', b') \\
&= R(s, b, a') + \gamma \sum_{s' \in S, b' \in B} P(s'|s, b, a')P(b'|s, b, a', s')V_\pi(s', b')
\end{aligned}
\tag{1}
$$

where the action is $a' = \pi(s, b)$.[1]

The Bayesian reward and transition functions are defined in expectation with respect to $\phi$: $R(s, b, a') = \sum_{\phi \in \Phi} b(\phi)R(s, \phi, a')$, $P(s'|s, b, a') = \sum_{\phi \in \Phi} b(\phi)P(s'|s, \phi, a')$. The belief distribution can be maintained recursively, with a black-box Bayes filter performing posterior updates given observations. We describe how to implement such a Bayes filter in Section 4.1.

The use of $(s, b)$ casts the partially observable BAMDP as a fully observable MDP in belief space, which permits the use of any policy gradient method. We highlight that a reactive Bayesian policy in belief space is equivalent to a policy with memory in observable space (Kaelbling et al., 1998). In our work, the complexity of memory is delegated to a Bayes filter that computes a sufficient statistic of the history.

In partially observable MDPs (POMDPs), the states can be observed only via a noisy observation function. Mixed-observability MDPs (MOMDPs) (Ong et al., 2010) are similar to BAMDPs: their states are $(s, \phi)$, where $s$ is observable and $\phi$ is latent. Although any BAMDP problem can be cast as a POMDP or a MOMDP problem (Duff & Barto, 2002), the source of uncertainty in a BAMDP usually comes from the transition function, not the unobservability of the state as it does with POMDPs and MOMDPs.

## 3 RELATED WORK

A long history of research addresses belief-space reinforcement learning and robust reinforcement learning. Here, we highlight the most relevant work and refer the reader to Ghavamzadeh et al. (2015), Shani et al. (2013), and Aberdeen (2003) for more comprehensive reviews of the Bayes-Adaptive and Partially Observable MDP literatures.

**Belief-Space Reinforcement Learning.** Planning in belief space, where part of the state representation is a belief distribution, is intractable (Papadimitriou & Tsitsiklis, 1987). This is a consequence of the curse of dimensionality: the dimensionality of belief space over a finite set of variables equals the size of that set, so the size of belief space grows exponentially. Many approximate solvers focus on one or more of the following: 1) value function approximation, 2) compact, approximate belief representation, or 3) direct mapping of belief to an action. QMDP (Littman et al., 1995) assumes full observability after one step to approximate Q-value. Point-based solvers, like SARSOP (Kurniawati et al., 2008) and PBVI (Pineau et al., 2003), exploit the piecewise-linear-convex structure of POMDP value functions (under mild assumptions) to approximate the value of a belief state. Sampling-based approaches, such as BAMCP (Guez et al., 2012) and POMCP (Silver & Veness, 2010), combine Monte Carlo sampling and simple rollout policies to approximate Q-values at the root node in a search tree. Except for QMDP, these approaches target discrete POMDPs and cannot be easily extended to continuous spaces. Sunberg & Kochenderfer (2018) extend POMCP to continuous spaces using double progressive widening. Model-based trajectory

---

[1] The state space $S$ can be either discrete or continuous. The belief space $B$ is always continuous, but we use $\sum$ notation for simplicity.

optimization methods (Platt et al., 2010; van den Berg et al., 2012) have also been successful for navigation on systems like unmanned aerial vehicles and other mobile robots.

Neural network variants of POMDP algorithms are well suited for compressing high-dimensional belief states into compact representations. For example, QMDP-Net (Karkus et al., 2017) jointly trains a Bayes-filter network and a policy network to approximate Q-value. Deep Variational Reinforcement Learning (Igl et al., 2018) learns to approximate the belief using variational inference and a particle filter, and it uses the belief to generate actions. Our method is closely related to Exp-GPOMDP (Aberdeen & Baxter, 2002), a model-free policy gradient method for POMDPs, but we leverage model knowledge from the BAMDP and revisit the underlying policy optimization method with recent advancements. Peng et al. (2018) use Long Short-Term Memory (LSTM) (Hochreiter & Schmidhuber, 1997) to encode a history of observations to generate an action. The key difference between our method and Peng et al. (2018) is that BPO explicitly utilizes the belief distribution, while in Peng et al. (2018) the LSTM must implicitly learn an embedding for the distribution. We believe that explicitly using a Bayes filter improves data efficiency and interpretability.

**Robust (Adversarial) Reinforcement Learning.**   One can bypass the burden of maintaining belief and still find a robust policy by maximizing the return for worst-case scenarios. Commonly referred to as Robust Reinforcement Learning (Morimoto & Doya, 2001), this approach uses a min-max objective and is conceptually equivalent to H-infinity control (Başar & Bernhard, 2008) from classical robust control theory. Recent works have adapted this objective to train agents against various external disturbances and adversarial scenarios (Pinto et al., 2017; Bansal et al., 2018; Pattanaik et al., 2018). Interestingly, instead of training against an adversary, an agent can also train to be robust against model uncertainty with an ensemble of models. For example, Ensemble Policy Optimization (EPOpt) (Rajeswaran et al., 2017) trains an agent on multiple MDPs and strives to improve worst-case performance by concentrating rollouts on MDPs where the current policy performs poorly. Ensemble-CIO (Mordatch et al., 2015) optimizes trajectories across a finite set of MDPs.

While adversarial and ensemble model approaches have proven to be robust even to unmodeled effects, they may result in overly conservative behavior when the worst-case scenario is extreme. In addition, since these methods do not infer or utilize uncertainty, they perform poorly when explicit information-gathering actions are required. Our approach is fundamentally different from them because it internally maintains a belief distribution. As a result, its policies outperform robust policies in many scenarios.

**Adaptive Policy Methods.**   Some approaches can adapt to changing model estimates without operating in belief space. Adaptive-EPOpt (Rajeswaran et al., 2017) retrains an agent with an updated source distribution after real-world interactions. PSRL (Osband et al., 2013) samples from a source distribution, executes an optimal policy for the sample for a fixed horizon, and then re-samples from an updated source distribution. These approaches can work well for scenarios in which the latent MDP is fixed throughout multiple episodes. Universal Policy with Online System Identification (UP-OSI) (Yu et al., 2017) learns to predict the maximum likelihood estimate $\phi_{MLE}$ and trains a universal policy that maps $(s, \phi_{MLE})$ to an action. However, without a notion of belief, both PSRL and UP-OSI can over-confidently execute policies that are optimal for the single estimate, causing poor performance in expectation over different MDPs.

## 4   Bayesian Policy Optimization

We propose Bayesian Policy Optimization, a simple policy gradient algorithm for BAMDPs (Algorithm 1). The agent learns a stochastic Bayesian policy that maps a state-belief pair to a probability distribution over actions $\pi : S \times B \to P(A)$. During each training iteration, BPO collects trajectories by simulating the current policy on several MDPs sampled from the prior distribution. During the simulation, the Bayes filter updates the posterior belief distribution at each timestep and sends the updated state-belief pair to the Bayesian policy. By simulating on MDPs with different latent variables, BPO observes the evolution of the state-belief throughout multiple trajectories. Since the state-belief representation makes the partially observable BAMDP a fully observable Belief-MDP, any batch policy optimization

---

**Algorithm 1** Bayesian Policy Optimization

---

**Require:** Bayes filter $\psi(\cdot)$, initial belief $b_0(\phi)$, $P_0$, policy $\pi_{\theta_0}$, horizon $H$, $n_{\text{itr}}, n_{\text{sample}}$

 1: **for** $i = 1, 2, \cdots, n_{\text{itr}}$ **do**
 2:     **for** $n = 1, 2, \cdots, n_{\text{sample}}$ **do**
 3:         Sample latent MDP $M$: $(s_0, \phi_0) \sim P_0$
 4:         $\tau_n \leftarrow \texttt{Simulate}(\pi_{\theta_{i-1}}, b_0, \psi, M, H)$
 5:     Update policy: $\theta_i \leftarrow \texttt{BatchPolicyOptimization}(\theta_{i-1}, \{\tau_1, \cdots, \tau_{n_{\text{sample}}}\})$
 6: **return** $\pi_{\theta_{best}}$

 7: **procedure** SIMULATE$(\pi, b_0, \psi, M, H)$
 8:     **for** $t = 1, \cdots, H$ **do**
 9:         $a_t \leftarrow \pi(s_{t-1}, b_{t-1})$
10:         Execute $a_t$ on $M$, observing $r_t, s_t$
11:         $b_t \leftarrow \psi(s_{t-1}, b_{t-1}, a_t, s_t)$
12:     **return** $(s_0, b_0, a_1, r_1, s_1, b_1, \cdots, a_H, r_H, s_H, b_H)$

---

algorithm (e.g., Schulman et al. (2015; 2017)) can be used to maximize the Bayesian Bellman equation (Equation 1).

One key challenge is how to represent the belief distribution over the latent state space. To this end, we impose one mild requirement, i.e., that the belief can be represented with a fixed-size vector. For example, if the latent space is discrete, we can represent the belief as a categorical distribution. For continuous latent state spaces, we can use Gaussian or a mixture of Gaussian distributions. When such specific representations are not appropriate, we can choose a more general uniform discretization of the latent space.

Discretizing the latent space introduces the curse of dimensionality. An algorithm must be robust to the size of the belief representation. To address the high-dimensionality of belief space, we introduce a new policy network structure that consists of two separate networks to independently encode state and belief (Figure 1b). These encoders consist of multiple layers of nonlinear (e.g., ReLU) and linear operations, and they output a compact representation of state and belief. We design the encoders to yield outputs of the same size, which we concatenate to form the input to the policy network. The encoder networks and the policy network are *jointly* trained by the batch policy optimization. Our belief encoder achieves the desired robustness by learning to compactly represent arbitrarily large belief representations. In Section 5, we empirically verify that the separate belief encoder makes our algorithm more robust to large belief representations (Figure 2b).

As with most policy gradient algorithms, BPO provides only a locally optimal solution. Nonetheless, it produces robust policies that scale to problems with high-dimensional observable states and beliefs (see Section 5).

### 4.1 BAYES FILTER FOR BAYESIAN POLICY OPTIMIZATION

Given an initial belief $b_0$, a Bayes filter recursively performs the posterior update:

$$b'(\phi'|s, b, a', s') = \eta \sum_{\phi \in \Phi} b(\phi) T(s, \phi, a', s', \phi') \tag{2}$$

where $\eta$ is the normalizing constant, and the transition function is defined as $T(s, \phi, a', s', \phi') = P(s', \phi'|s, \phi, a') = P(s'|s, \phi, a') P(\phi'|s, \phi, a', s')$. At timestep $t$, the belief $b_t(\phi_t)$ is the posterior distribution over $\Phi$ given the history of states and actions, $(s_0, a_1, s_1, ..., s_t)$. When $\phi$ corresponds to physical parameters for an autonomous system, we often assume that the latent states are fixed.

Our algorithm utilizes a black-box Bayes filter to produce a posterior distribution over the latent states. Any Bayes filter that outputs a fixed-size belief representation can be used; for example, we use an extended Kalman filter to maintain a Gaussian distribution over continuous latent variables in the LightDark environment in Section 5. When such a specific

representation is not appropriate, we can choose a more general discretization of the latent space to obtain a computationally tractable belief update.

For our algorithm, we found that uniformly discretizing the range of each latent parameter into $K$ equal-sized bins is sufficient. From each of the resulting $K^{|\Phi|}$ bins, we form an MDP by selecting the mean bin value for each latent parameter. Then, we approximate the belief distribution with a categorical distribution over the resulting MDPs.

We approximate the Bayes update in Equation 2 by computing the probability of observing $s'$ under each discretized $\phi \in \{\phi_1, \cdots, \phi_{K^{|\Phi|}}\}$ as follows:

$$b'(\phi|s, b, a', s') = \frac{b(\phi)p(s'|s, \phi, a')}{\sum_{i=1}^{K^{|\Phi|}} b(\phi_i)p(s'|s, \phi_i, a')}$$

where the denominator corresponds to $\eta$.

As we verify in Section 5, our algorithm is robust to approximate beliefs, which allows the use of computationally efficient approximate Bayes filters without degrading performance. A belief needs only to be accurate enough to inform the agent of its actions.

## 4.2 Generalization to POMDP

Although BPO is designed for BAMDP problems, it can naturally be applied to POMDPs. In a general POMDP where state is unobservable, we need only $b(s)$, so we can remove the state encoder network.

Knowing the transition and observation functions, we can construct a Bayes filter that computes the belief $b$ over the hidden state:

$$b'(s') = \psi(b, a', o') = \eta \sum_{s \in S} b(s)T(s, a', s')Z(s, a', o')$$

where $\eta$ is the normalization constant, and $Z$ is the observation function, $Z(s, a', o') = P(o'|s, a')$, of observing $o'$ after taking action $a'$ at state $s$. Then, BPO optimizes the following Bellman equation:

$$V_\pi(b) = \sum_{s \in S} b(s)R(s, \pi(b)) + \gamma \sum_{b' \in B} P(b'|b, \pi(b))V_\pi(b')$$

For general POMDPs with large state spaces, however, discretizing state space to form the belief state is impractical. We believe that this generalization is best suited for beliefs with conjugate distributions, e.g., Gaussians.

## 5 Experimental Results

We evaluate BPO on discrete and continuous POMDP benchmarks to highlight its use of information-gathering actions. We also evaluate BPO on BAMDP problems constructed by varying physical model parameters on OpenAI benchmark problems (Brockman et al., 2016). For all BAMDP problems with continuous latent spaces (`Chain`, `MuJoCo`), latent parameters are sampled in the continuous space in Step 3 of Algorithm 1, regardless of discretization.

We compare BPO to EPOpt and UP-MLE, robust and adaptive policy gradient algorithms, respectively. We also include BPO-, a version of our algorithm without the belief and state encoders; this version directly feeds the original state and belief to the policy network. Comparing with BPO- allows us to better understand the effect of the encoders. For UP-MLE, we use the maximum likelihood estimate (MLE) from the same Bayes filter used for BPO, instead of learning an additional online system identification (OSI) network as originally proposed by UP-OSI. This lets us directly compare performance when a full belief distribution is used (BPO) rather than a point estimate (UP-MLE). For the OpenAI BAMDP problems, we also compare to a policy trained with TRPO in an environment with the mean values of the latent parameters.

All policy gradient algorithms (BPO, BPO-, EPOpt, UP-MLE) use TRPO as the underlying batch policy optimization subroutine. We refer the reader to Appendix A.1 for

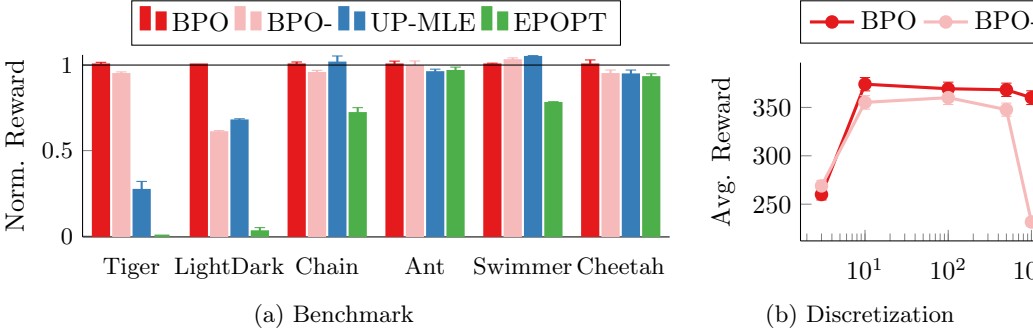

(a) Benchmark          (b) Discretization

Figure 2: (a) Comparison of BPO with belief-agnostic, robust RL algorithms. BPO significantly outperforms benchmarks when belief-awareness and explicit information gathering are necessary (`Tiger`, `LightDark`). It is competitive with UP-MLE when passive estimation or universal robustness is sufficient (`Chain`, `MuJoCo`). (b) Scalability of BPO with respect to latent state space discretization for the `Chain` problem.

parameter details. For all algorithms, we compare the results from the seed with the highest mean reward across multiple random seeds. Although EPOpt and UP-MLE are the most relevant algorithms that use batch policy optimization to address model uncertainty, we emphasize that neither formulates the problems as BAMDPs.

As shown in Figure 1b, the BPO network's state and belief encoder components are identical, consisting of two fully connected layers with $N_h$ hidden units each and tanh activations ($N_h = 32$ for `Tiger`, `Chain`, and `LightDark`; $N_h = 64$ for `MuJoCo`). The policy network also consists of two fully connected layers with $N_h$ hidden units each and tanh activations. For discrete action spaces (`Tiger`, `Chain`), the output activation is a softmax, resulting in a categorical distribution over the discrete actions. For continuous action spaces (`LightDark`, `MuJoCo`), we represent the policy as a Gaussian distribution.

Figure 2a illustrates the normalized performance for all algorithms and experiments. We normalize by dividing the total reward by the reward of BPO. For `LightDark`, which has negative reward, we first shift the total reward to be positive and then normalize. Appendix A.2 shows the unnormalized rewards.

**Tiger (Discrete POMDP).** In the `Tiger` problem, originally proposed by Kaelbling et al. (1998), a tiger is hiding behind one of two doors. An agent must choose among three actions: listen, or open one of the two doors; when the agent listens, it receives a noisy observation of the tiger's position. If the agent opens the door and reveals the tiger, it receives a penalty of -100. Opening the door without the tiger results in a reward of 10. Listening incurs a penalty of -1. In this problem, the optimal agent listens until its belief about which door the tiger is behind is substantially higher for one door vs. the other. Chen et al. (2016) frame `Tiger` as a BAMDP problem with two latent states, one for each position of the tiger.

Figure 2a demonstrates the benefit of operating in state-belief space when information gathering is required to reduce model uncertainty. Since the EPOpt policy does not maintain a belief distribution, it sees only the most recent observation. Without the full history of observations, EPOpt learns only that opening doors is risky; because it expects the worst-case scenario, it always chooses to listen. UP-MLE leverages all past observations to estimate the tiger's position. However, without the full belief distribution, the policy cannot account for the confidence of the estimate. Once there is a higher probability of the tiger being on one side, the UP-MLE policy prematurely chooses to open the safer door. BPO significantly outperforms both of these algorithms, learning to listen until it is extremely confident about the tiger's location. In fact, BPO achieves close to the approximately optimal return found by SARSOP ($19.0 \pm 0.6$), a state-of-the-art offline POMDP solver that approximates the optimal value function rather than performing policy optimization (Kurniawati et al., 2008).

|  | BPO | BEETLE | PERSEUS | MCBRL |
|---|---|---|---|---|
| Chain-10 (tied) | $364.5 \pm 0.5$ | $365.0 \pm 0.4$ | $366.1 \pm 0.2$ | |
| Chain-10 (semitied) | $364.9 \pm 0.8$ | $364.8 \pm 0.3$ | $365.1 \pm 0.3$ | $321.6 \pm 6.4$ |

Table 1: For the `Chain` problem, a comparison of the 95% confidence intervals of average return for BPO vs. other benchmark algorithms. Values for BEETLE, MCBRL, and Perseus are taken from Wang et al. (2012), which does not report MCBRL performance in the "tied" setting.



Figure 3: Visualization of different algorithms on the `LightDark` environment. The dashed line indicates the light source. Blue circles are one standard deviation for per-step estimates. The BPO policy moves toward the light to obtain a better state estimate before moving toward the goal.

**Chain (Discrete BAMDP).** To evaluate the usefulness of the independent encoder networks, we consider a variant of the `Chain` problem (Strens, 2000). The original problem is a discrete MDP with five states $\{s_i\}_{i=1}^{5}$ and two actions $\{A, B\}$. Taking action $A$ in state $s_i$ transitions to $s_{i+1}$ with no reward; taking action $A$ in state $s_5$ transitions to $s_5$ with a reward of 10. Action $B$ transitions from any state to $s_1$ with a reward of 2. However, these actions are noisy: in the canonical version of `Chain`, the opposite action is taken with slip probability 0.2. In our variant, the slip probability is uniformly sampled from $[0, 1.0]$ at the beginning of each episode.[2] In this problem, either action provides equal information about the latent parameter. Since active information-gathering actions do not exist, BPO and UP-MLE achieve similar performance.

Figure 2b shows that our algorithm is robust to the size of latent space discretization. We discretize the parameter space with 3, 10, 100, 500, and 1000 uniformly spaced samples. At coarser discretizations (3, 10), we see little difference between BPO and BPO-. However, with a large discretization (500, 1000), the performance of BPO- degrades significantly, while BPO maintains comparable performance. The performance of BPO also slightly degrades when the discretization is too fine, suggesting that this level of discretization makes the problem unnecessarily complex. Figure 2a shows the best discretization (10).

In this discrete domain, we compare BPO to BEETLE (Poupart et al., 2006) and MCBRL (Wang et al., 2012), state-of-the-art discrete Bayesian reinforcement learning algorithms, as well as Perseus (Spaan & Vlassis, 2005), a discrete POMDP solver. In addition to our variant, we consider a more challenging version where the slip probabilities for both actions must be estimated independently. Poupart et al. (2006) refer to this as the "semi-tied" setting; our variant is "tied." BPO performs comparably to all of these benchmarks (Table 1).

**Light-Dark (Continuous POMDP).** We consider a variant of the `LightDark` problem proposed by Platt et al. (2010), where an agent tries to reach a known goal location while being uncertain about its own position. At each timestep, the agent receives a noisy observation of its location. In our problem, the vertical dashed line is a light source; the farther the agent is from the light, the noisier its observations. The agent must decide either to reduce uncertainty by moving closer to the light, or to exploit by moving from its estimated position to the goal. We refer the reader to Appendix A.3 for details about the rewards and observation noise model.

---

[2] A similar variant was introduced in Wang et al. (2012).

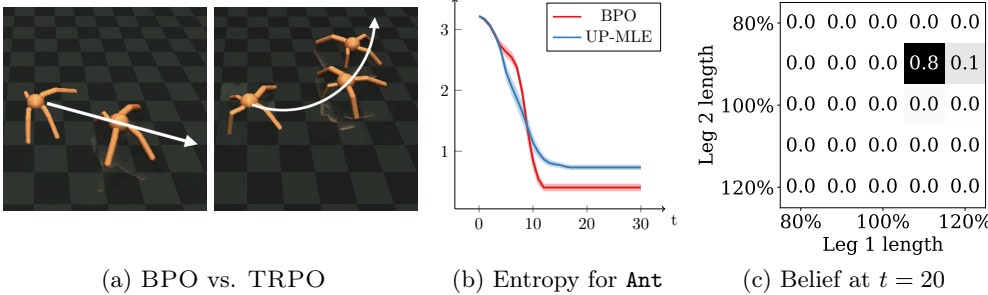

(a) BPO vs. TRPO      (b) Entropy for `Ant`      (c) Belief at $t = 20$

Figure 4: (a) Comparison of BPO and TRPO trained on the nominal environment for a different environment. The task is to move to the right along the x-axis. However, the model at test time differs from the one TRPO trained with: one leg is 20% longer, another is 20% shorter. (b) Comparison of average entropy per timestep by BPO and UP-MLE. The belief distribution collapses more quickly under the BPO policy. (c) Belief distribution at $t = 20$ during a BPO rollout.

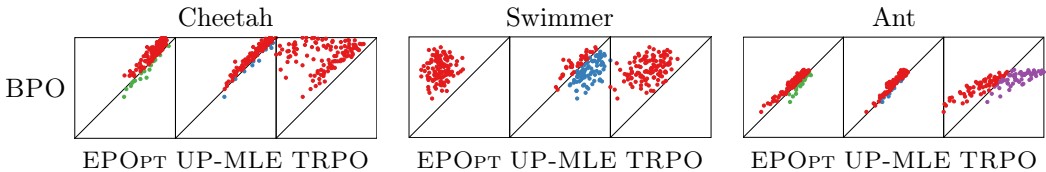

Figure 5: Pairwise performance comparison of algorithms on `MuJoCo` BAMDPs. Each point represents an MDP, and its $(x, y)$-coordinates correspond to the long-term reward by (baseline, BPO). The farther a point is above the line $y = x$, the more BPO outperforms that baseline. Colors indicate which algorithm achieved higher reward: BPO (red), EPOPT (green), UP-MLE (blue), or TRPO (purple).

This example demonstrates how to apply BPO to general continuous POMDPs (Section 4.2). The latent state is the continuous pose of the agent. For this example, we parameterize the belief as a Gaussian distribution and perform the posterior update with an Extended Kalman Filter, as in Platt et al. (2010).

Figure 3 compares sample trajectories from different algorithms on the `LightDark` environment. Based on its initial belief, the BPO policy moves toward a light source to acquire less noisy observations. As it becomes more confident in its position estimate, it changes direction toward the light and then moves straight to the goal. Both EPOPT and UP-MLE move straight to the goal without initially reducing uncertainty.

**MuJoCo (Continuous BAMDP).** Finally, we evaluate the algorithms on three simulated benchmarks from OpenAI Gym (Brockman et al., 2016) using the MuJoCo physics simulator (Todorov et al., 2012): `HalfCheetah`, `Swimmer`, and `Ant`. Each environment has several latent physical parameters that can be changed to form a BAMDP. We refer the reader to Appendix A.4 for details regarding model variation and belief parameterization.

The MuJoCo benchmarks demonstrate the robustness of BPO to model uncertainty. For each environment, BPO learns a universal policy that adapts to the changing belief over the latent parameters.

Figure 4 highlights the performance of BPO on `Ant`. BPO can efficiently move to the right even when the model substantially differs from the nominal model (Figure 4a). It takes actions that reduce entropy more quickly than UP-MLE (Figure 4b). The belief over the possible MDPs quickly collapses into a single bin (Figure 4c), which allows BPO to adapt the policy to the identified model.

Figure 5 provides a more in-depth comparison of the long-term expected reward achieved by each algorithm. In particular, for the `HalfCheetah` environment, BPO has a higher average return than both EPOPT and UP-MLE for most MDPs. Although BPO fares slightly worse

than UP-MLE on `Swimmer`, we believe that this is largely due to random seeds, especially since BPO- matches UP-MLE's performance (Figure 2a).

Qualitatively, all three algorithms produced agents with reasonable gaits in most MDPs. We postulate two reasons for this. First, the environments do not require active information-gathering actions to achieve a high reward. Furthermore, for deterministic systems with little noise, the belief collapses quickly (Figure 4b); as a result, the MLE is as meaningful as the belief distribution. As demonstrated by Rajeswaran et al. (2017), a universally robust policy for these problems is capable of performing the task. Therefore, even algorithms that do not maintain a history of observations can perform well.

## 6  Discussion

Bayesian Policy Optimization is a practical and scalable approach for continuous BAMDP problems. We demonstrate that BPO learns policies that achieve performance comparable to state-of-the-art discrete POMDP solvers. They also outperform state-of-the-art robust policy gradient algorithms that address model uncertainty without formulating it as a BAMDP problem. Our network architecture scales well with respect to the degree of latent parameter space discretization due to its independent encoding of state and belief. We highlight that BPO is agnostic to the choice of batch policy optimization subroutine. Although we used TRPO in this work, we can also use more recent policy optimization algorithms, such as PPO (Schulman et al., 2017), and leverage improvements in variance-reduction techniques (Weaver & Tao, 2001).

BPO outperforms algorithms that do not explicitly reason about belief distributions. Our Bayesian approach is necessary for environments where uncertainty must actively be reduced, as shown in Figure 2a and Figure 3. If all actions are informative (as with `MuJoCo`, `Chain`) and the posterior belief distribution easily collapses into a unimodal distribution, UP-MLE provides a lightweight alternative.

BPO scales to fine-grained discretizations of latent space. However, our experiments also suggest that each problem has an optimal discretization level, beyond which further discretization may degrade performance. As a result, it may be preferable to perform variable-resolution discretization rather than an extremely fine, single-resolution discretization. Adapting iterative densification ideas previously explored in motion planning (Gammell et al., 2015) and optimal control (Munos & Moore, 1999) to the discretization of latent space may yield a more compact belief representation while enabling further improved performance.

An alternative to the model-based Bayes filter and belief encoder components of BPO is learning to directly map a history of observations to a lower-dimensional belief embedding, analogous to Peng et al. (2018). This would enable a policy to learn a meaningful belief embedding without losing information from our a priori choice of discretization. Combining a recurrent policy for unidentified parameters with a Bayes filter for identified parameters offers an intriguing future direction for research efforts.

### Acknowledgments

Gilwoo Lee is partially supported by Kwanjeong Educational Foundation, and Brian Hou is partially supported by NASA Space Technology Research Fellowships (NSTRF). This work was partially funded by the National Institute of Health R01 (#R01EB019335), National Science Foundation CPS (#1544797), National Science Foundation NRI (#1637748), the Office of Naval Research, the RCTA, Amazon, and Honda.

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

## Appendix

### A.1 Training Parameters

The encoder networks and policy network are jointly trained with Trust Region Policy Optimization (Schulman et al., 2015). We used the implementation provided by Duan et al. (2016) with the parameters listed in Appendix Table 1.

|  | Tiger | Chain | LightDark | MuJoCo |
|---|---|---|---|---|
| Max. episode length | 100 | 100 | 15 | 200 |
| Batch size | 500 | 10000 | 400 | 500 |
| Training iterations | 1000 | 500 | 10000 | 200 |
| Discount ($\gamma$) | 0.95 | 1.00 | 1.00 | 0.99 |
| Stepsize ($\overline{D}_{KL}$) | 0.01 | 0.01 | 0.01 | 0.01 |
| GAE $\lambda$ | 0.96 | 0.96 | 0.96 | 0.96 |

Appendix Table 1: Training parameters

### A.2 Unnormalized Experimental Results

Here, we provide unnormalized experimental results for the normalized performance in Figure 2a.

|  | BPO | BPO- | EPOpt | UP-MLE | TRPO |
|---|---|---|---|---|---|
| Tiger | **17.9** $\pm$ 0.6 | 15.8 $\pm$ 0.6 | -19.9 $\pm$ 0.0 | -9.8 $\pm$ 2.0 | - |
| Chain-3 | **260.1** $\pm$ 5.6 | **268.9** $\pm$ 5.7 | **267.9** $\pm$ 13.1 | 242.0 $\pm$ 11.2 | - |
| Chain-10 | **374.0** $\pm$ 6.9 | 355.2 $\pm$ 7.0 | 267.9 $\pm$ 13.1 | **378.2** $\pm$ 15.7 | - |
| Chain-1000 | **360.1** $\pm$ 7.1 | 231.6 $\pm$ 4.2 | 267.9 $\pm$ 13.1 | **342.4** $\pm$ 14.9 | - |
| LightDark | **-166.7** $\pm$ 2.4 | -867.9 $\pm$ 22.1 | -1891.2 $\pm$ 45.0 | -745.9 $\pm$ 22.3 | - |
| HalfCheetah | **115.6** $\pm$ 3.5 | 109.13 $\pm$ 3.1 | 107.0 $\pm$ 2.7 | 108.9 $\pm$ 3.3 | 64.3 $\pm$ 6.1 |
| Swimmer | 36.0 $\pm$ 0.4 | **36.9** $\pm$ 0.6 | 27.9 $\pm$ 0.4 | **37.6** $\pm$ 0.4 | 29.4 $\pm$ 0.6 |
| Ant | **117.0** $\pm$ 2.7 | 115.9 $\pm$ 3.9 | 112.5 $\pm$ 3.1 | 111.7 $\pm$ 2.5 | **116.5** $\pm$ 7.5 |

Appendix Table 2: Comparison of the 95% confidence intervals of average return for BPO and other benchmark algorithms across all environments. Algorithms with the highest average return on each environment are shown in bold, with multiple algorithms selected if intervals overlap. BPO achieves the highest return on seven of the eight environments. The combined result of BPO and BPO- achieves the highest return on all environments.

### A.3   EXPERIMENTAL DETAIL: LIGHTDARK

After each action, an agent receives a noisy observation of its location, which is sampled from a Gaussian distribution, $o \sim \mathcal{N}([x, y]^\top, w(x))$, where $[x, y]$ is the true location. The noise variance is a function of $x$ and is minimized when $x = 5$: $w(x) = \frac{1}{2}(x - 5)^2 + \text{const}$. There is no process noise.

The reward function is $r(s, a) = -\frac{1}{2}(\|s - g\|^2 + \|a\|^2)$, where $s$ is the true agent position and $g$ is the goal position. A large penalty of $-5000\|s_T - g\|^2$ is incurred if the agent does not reach the goal by the end of the time horizon, analogous to the strict equality constraint in the original optimization problem (Platt et al., 2010).

The initial belief is $[x, y, \sigma^2] = [2, 2, 2.25]$. During training, we randomly sample latent start positions from a rectangular region $[2, -2] \times [4, 4]$ and observable goal positions from $[0, -2] \times [2, 4]$.

### A.4   EXPERIMENTAL DETAIL: MUJOCO

For ease of analysis, we vary two parameters for each environment. For `HalfCheetah`, the front and back leg lengths are varied. For `Ant`, the two front leg lengths are varied. `Swimmer` has four body links, so the first two link lengths vary together according to the first parameter, and the last two links vary together according to the second parameter. We chose to vary link lengths rather than friction or the damping constant because a policy trained on a single nominal environment can perform well across large variations in those parameters. All link lengths vary by up to 20% of the original length.

To construct a Bayes filter, the 2D-parameter space is discretized into a $5 \times 5$ grid with a uniform initial belief. We assume Gaussian noise on the observation, i.e. $o = f_\phi(s, a) + w$ with $w \sim \mathcal{N}(0, \sigma^2)$, with $\phi$ being the parameter corresponding to the center of each grid cell. It typically requires only a few steps for the belief to concentrate in a single cell of the grid, even when a large $\sigma^2$ is assumed.

