# OpenReview forum: "Bayesian Policy Optimization for Model Uncertainty"
_ICLR.cc/2019/Conference_

### Official Review · AnonReviewer1 · 2018-11-02
**Solid**

**Rating:** 7
**Confidence:** 3

**Review:**

Evaluation:
This is a solid paper: The idea is clear, it is well communicated and put into context of the existing literature, and the results are promising. The experiments are well chosen and illustrate the method well. The connection between the chosen setting (BAMDPs) to POMDPs is explained well and explored in the empirical evaluation as well. I think that the methods section could go into a bit more detail, and the underlying assumptions that the authors make could be discussed more critically.

Summary:
This paper looks at Bayes-Adaptive MDPs (BAMDPs) in which the latent parameter space is either
- a discrete finite set or
- a bounded continuous set that can be approximated via discretization.
Consequently, the authors choose to represent the belief as a categorical distribution, which can be represented by a vector of weights.
They further assume that the environment model is known. Hence, the posterior belief can be computed exactly.
If I understand correctly, the main contribution is that the authors represent the policy as a neural network and train it using a policy gradient algorithm.
This is a good first step towards scalable Bayesian policy optimisation.

Main Feedback:
- In the Introduction, first paragraph, you say one of the aspects of real-world robotics is that there's "(1) an underlying dynamical system with unknown latent parameters". I would argue that the dynamic system itself is typically also unknown, including how it is parametrized by these latent parameters. I think it is important to point this out more explicitly in the introduction (it is mentioned in sec 2 and 5, but maybe it's worth mentioning it in 4 again as well): for the problems that you look at, you assume that the form of the transition function is known (just not its parameters phi).
- In the main methods section (4), it would be nice to see some more detail about the Bayes filter. Can you write out the distribution over the latent parameters, and write out how the filtering is done? Explain how to compute the normalising constant (and mention explicitly why this is possible for your set-up, and why it would be infeasible if the latent space cannot be discretized). How exactly is the posterior distribution represented and fed to the policy? Seeing this done explicitly in Section 4 (even if it repeats some things that are explained in 2) would help someone that is interested in (re-)implementing the proposed method.
- I would like to see a more critical discussion in Section 7 about the assumptions that the authors make: that the environment models are known, and that the latent space can be discretized. How realistic are those assumptions (and in which kind of real-world problems can we make them), and what are ways forward to drop these assumptions?

Other Comments:
- Introduction: Using an encoder for the state/belief is an implementation choice, and (as I see it) not part of the main contribution. I would focus on explaining the intuition behind BPO in the introduction, and only mention the architecture choice as a side note.
- Related Work: The authors might be interested in the recent work of Igl et al. (ICML 2018, "Deep Variational RL for POMDPs"), who approximate the belief in a POMDP using variational inference and a particle filter.

Significance for ICLR:
- In the light-dark experiment, the authors visualise the belief that the agent has at every time step. It would have been nice to see an analysis of how exactly the belief looks also for maybe 1-2 other experiments, and how (when) the agent makes a decision based on this. This could replace Table 2 (which I guess should be called Figure 2?), which I did not find very insightful.

---

> ### Author Response · Authors · 2018-11-25
> **Revision addresses the comments; additional clarification below**
>
> Thank you for your feedback. In addition to the clarification on belief representation (Section 4), the newly added section on Bayes filter (Section 5), and newly added figures (Figure 2 and 4) based on your suggestions, we would like to answer some of the concerns you have raised.
>
>
> Regarding the assumption that the environment models are known and/or that the latent space can be discretized:
>
> We believe that there are largely three classes of robotics problems where BAMDPs can be applied:
> 1) The real-world dynamics can be reasonably approximated by simulators (or closed-form dynamics equations).
> 2) Although the simulator differs from the real-world dynamics, leveraging domain randomization while training a robust policy successfully transfers to the real world.
> 3) The real-world dynamics must be learned from scratch (possibly in a nonparametric manner).
>
> Many existing approaches in robust RL aim for 1 and 2. Some example scenarios are varying spring coefficients modeling the wear and tear of robotic legs or manipulation of a set of objects whose mass or shape can be parametrized. In these cases, we believe that our approach will produce better policies than belief-agnostic approaches.
>
> For case 3, we would like to note that our algorithm can in fact learn a policy even when there is little prior dynamics knowledge from simulators. For discrete BAMDPs, independent Dirichlet distributions for p(s’|s, a) are a common choice for uninformative priors (Duff & Barto, 2002; Kolter & Ng, 2009). For continuous space transition functions, we can maintain a joint distribution of continuous random variables e.g. Gaussian processes. Using these flexible priors requires no change to Algorithm 1, although the step of sampling an MDP during training would now involve sampling from the Dirichlet or Gaussian process. In the case of GP, the input to the policy network has to be a fixed-size representation of the GP posterior distribution. This would be an interesting future work.
>
> In summary, our algorithm can leverage prior knowledge from simulators (cases 1 and 2), but this does not limit it from learning almost from scratch (case 3).

---

> > ### Comment · AnonReviewer1 · 2018-11-27
> > **Reply**
> >
> >
> > I looked at the revised paper, and think the authors did a good job at incorporating our feedback. I particularly like Section 5.
> >
> > Figure 4 in general is interesting, but it's still difficult to infer how beliefs change over time or how the agent makes decisions based on it (which, admittedly, is not easy to show on paper - if the authors ever have time I highly recommend making a video with the ant moving and the heat map next to it :))
> >
> > In 4a, is the +x direction parallel to the lower edge of the image? How did the task change? The figure can potentially give the reader some insight into how the policies adapt given some environment change; however I think some details are missing.

---

> > > ### Author Response · Authors · 2018-12-03
> > > **Thank you for suggested changes**
> > >
> > > Thank you for reading the revised paper and suggesting further changes. In the next version, we will submit a supplementary video with the ant and the heat map evolution.
> > >
> > > Yes, +x is the direction parallel to the lower edge. The task remains the same, i.e. to move toward +x, but the ant model is different from the nominal model (used for TRPO) in that one of the legs is 20% longer and another is 20% shorter. We will include this detail in the next draft.

---

### Official Review · AnonReviewer2 · 2018-11-02
**Experiment results are not much convincing.**

**Rating:** 7
**Confidence:** 4

**Review:**

Summary: This paper proposes a policy optimization framework for Bayesian RL (BPO). BPO is based on a Bayesian model-based RL formulation. Using a Bayesian approach, it is expected to have better trade-off between exploration and exploitation in RL, and be able to deal with model uncertainty as well. Experiments are done on multiple domains consisting both POMDP planning tasks and RL.

In general, the paper is well written. Related work are thoroughly discussed. In my opinion, the proposed idea is a solid combination of existing techniques: Monte-Carlo sampling (step 3), Bayes belief update, and policy gradient in POMDP (G(PO)MDP). However, this combination is still worth trying and has been shown to scale to larger problems through the use of deep learning.

I have some following major concerns about the paper:

- Root sampling (step 3 in Algorithm 1) would result in sampled models that are fixed in every simulation. In a pure nature of Bayes RL, after each update at new observation (step 11: belief update), the model distribution already changes. Thus how does this Algorithm can guarantee an optimal solution for BAMDP? can the authors have more discussions on this point? Does this explain why TRPO (using a mean model) can perform comparably to BPO in Ant?

- Belief representation is based on a Bayes filter which requires discretization. Finely discretized belief would increase the complexity and computation dramatically with the dimension of the latent space. This would result in very slow SIMULATE steps, especially for a long-horizon problem, let alone further computation for BatchPolicyOptimization.

- I wonder how TRPO using RNN would perform in this case, instead of using a wrong starting model (an average model)?

---

> ### Author Response · Authors · 2018-11-25
> **Revision addresses the comments; additional clarification below**
>
> Thank you for your feedback. In addition to the clarification on belief representation (Section 4) and the newly added section on Bayes filter (Section 5), we would like to answer some of the concerns you have raised.
>
> Root sampling results in sampled models that are fixed in every simulation:
> This is indeed the correct realization of the BAMDP framework, where the underlying model is fixed but unknown. Our algorithm addresses this by fixing the sampled model for the whole episode. Since the true model is hidden from the agent, it maintains a belief over the possible models. After each belief update, the agent’s belief over the model changes, but the actual underlying model remains the same. A Bayes-optimal agent learns to act such that the uncertainty in the belief distribution reduces to the degree necessary for maximal long-term reward.
>
> TRPO on Ant performs well on certain cases but poorly on corner cases. The reason why BPO seems to have only marginal gain in this case is due to the particular four-legged nature of Ant, which allows a mean-model agent to walk reasonably under small geometric variation. The visualization of a corner case is added in Figure 4.
>
>
> Computational complexity of discretization:
> We agree that increasing the belief discretization level increases the time required to perform posterior updates at each timestep. Ultimately, this is an implementation detail of the black-box Bayes filter. However, we have empirically found that fine discretization of the continuous latent state space may be unnecessary: BPO produces high-performing agents even with a coarse discretization. For MuJuCo problems, we outperform the other baselines with only 25 bins to discretize the latent parameter space. For the Chain problem, the discretization with 10 bins is as good as or slightly better than 1e2 or 1e3 bins. This implies two things: 1) our algorithm is robust to approximate beliefs, and 2) the agent only needs the belief to be sufficiently accurate to inform its actions. Due to these properties, we believe that more computationally-efficient approximate Bayes filters can be used without significantly degrading performance.
>
>
> RNN:
> As you suggest, a recurrent policy could learn to act with respect to a history of observations. In our case, the history of observation is encoded by the belief, so TRPO in the belief space has as much information as an RNN. The use of RNN for jointly training the Bayes filter and the policy could certainly be effective, as proposed in (Karkus et al., 2017).

---

### Official Review · AnonReviewer3 · 2018-11-05

**Rating:** 6
**Confidence:** 3

**Review:**

In this paper, the author proposed to utilize a novel policy structure and recent batch policy optimization methods such as PPO or TRPO to solve Bayes-Adaptive MDP (BAMDP) and Partial Observable MDP(POMDP) problems. The author verified the proposed method on discrete and continuous POMDP and BAMDP benchmarks compared with other baseline methods.

The main part of the paper is trying to explain Bayesian RL and the relationship between BAMDP and POMDP, and several related work. There is only a half page that explains the main idea of the proposed method, and it seems that the author combines several existing techniques and utilize deep learning to solve BAMDP and POMDP problems.

The detail of the experiment is not clarified explicitly, such as the structure and size of the policy, training details of the BPO, and detail parameters changed to formulate BAMDP for Mujoco environments.

The paper strikes me as a valuable contribution once the detail of the experiments are addressed, but personally I am not sure that whether the novelty of this paper is enough for the main conference track.

---

> ### Author Response · Authors · 2018-11-25
> **Revision includes experimental details.**
>
> Thank you for your feedback.
>
> We have added experimental details regarding the policy network (Section 6) and training parameters (Appendix A3). The model parameter ranges for MuJoCo BAMDP problems are described in Appendix A2.

---

### Official Review · AnonReviewer4 · 2018-11-29
**Combination of several existing methods + none quite convincing experiments**

**Rating:** 5
**Confidence:** 4

**Review:**

Summary:

In this paper, the authors propose a policy gradient algorithm for solving a Bayes-Adaptive MDP (BAMDP). At each iteration, the algorithm samples several MDPs from the prior distribution and simulates a trajectory for each sampled MDP. During the simulation, the algorithm uses a Bayes filter to update the posterior belief distribution at each time step. Finally, the algorithm uses the sampled trajectories and update the policy using the TRPO algorithm.

The authors propose to pass the state and belief through separate encoders, to reduce their dimensions, and then put them together and give them to the policy network. Although the experiments show that the encoding did not improve the performance significantly, except in the Lightdark problem.

The authors show that their algorithm can also be used to solve POMDPs by replacing the state-belief pair with just belief. Basically turning a POMDP to a belief state MDP and then applying the algorithm. They evaluate their algorithm in two POMDP problems, one discrete and one continuous, in both their algorithm achieves a reasonable performance.

A tricky part of the algorithm is how to define a Bayes filter for continuous latent states. This is crucial in updating the posterior after each observation. The way the authors handle this is by discretization, and how the discretization should be done (high or low resolution) is a hyper parameter. Although the experiments indicate that the proposed algorithm, especially with encoders, is quite robust w.r.t. the discretization.


Comments:

- The idea behind the algorithm proposed in the paper is quite simple. It is a combination of Bayesian optimization (sampling several MDPs from the prior), using a Bayes filter to update the belief, and a policy gradient algorithm (TRPO) to estimate the gradient and update the policy parameter. The only challenges are 1) the design of the Bayes filter, in particular when the latent state is continuous, in which the idea used in the paper is very simple, discretization, and 2) dealing with potentially high dimensional state-belief pair, which was handled by the encoders.

- The structure of the paper could be improved significantly. Four pages have been dedicated to the preliminaries and related work, and another four pages to the experiments. This leaves only less than two pages for the algorithm. While I think a comprehensive discussion of the experiments is quite helpful, I found the preliminaries and related work too long. I even think that the experiments could have been written better. There are parts that have been explained too much and parts that are not clear or left for the appendix. With a better structure, the algorithm could have been explained better. I personally would like to see more discussion on how the distribution over MDPs is updated.

-  I did not find the experiments very convincing. In BAMDP problems (Chain and MuJoCos), the proposed algorithm performs similarly to the adaptive policy gradient method. We only see improvement in the POMDP tasks (Tiger, Lightdark), which I think the main reason is that the algorithms selected for comparison are not the right algorithms for POMDPs. For example, many different algorithms have been used to solve Tiger (or other discrete POMDPs) in the POMDP literature, and I do not see any of them in the paper.

---

> ### Author Response · Authors · 2018-12-03
> **Reply**
>
> Thank you for your feedback. We believe that policy optimization is a promising direction for solving continuous Bayesian Reinforcement Learning problems which has not been sufficiently explored. Reviewer 2 agrees that the proposed approach is a "solid combination of existing techniques....[that] is still worth trying and has been shown to scale to larger problems through the use of deep learning."
>
> BPO performs similarly to the adaptive policy gradient method (UP-MLE) on Chain and MuJoCo BAMDP problems. As we observe in Section 8, "if all actions are informative (as with MuJoCo, Chain) and the posterior belief distribution easily collapses into a unimodal distribution, UP-MLE provides a lightweight alternative." However, BPO significantly outperforms UP-MLE in the Tiger and LightDark domains, which are POMDP problems where not all actions are informative. The success in POMDP problems highlights that the BPO agent is indeed belief-aware. BPO generalizes to POMDPs, which we discuss in Section 6.
>
> EPOpt and UP-MLE are not Bayesian algorithms: "Although EPOpt and UP-MLE are the most relevant algorithms which utilize batch policy optimization to address model uncertainty, we emphasize that neither formulate the problems as BAMDPs" (Section 7).  However, as we point out in Section 2, many state-of-the-art BAMDP or POMDP solvers are designed for discrete state-action spaces, which cannot be readily extended to continuous spaces (e.g. the MuJoCo domain). We have focused on comparisons with algorithms that deal with continuous spaces, while providing POMDP baselines for context wherever possible. In the Tiger domain, BPO nearly matches the performance of SARSOP (Kurniawati et al., 2008). The BPO trajectory on LightDark nearly matches the optimal trajectory drawn in Figure 1 of (Platt et al., 2010). In addition, we have run further experiments that compare BPO with other discrete Bayesian reinforcement learning algorithms (BRL) on the Chain domain (Poupart et al., 2006), which we discuss below. These experiments validate that our BPO performs as well as state-of-the-art BRL or POMDP solvers in discrete domains, while opening a promising direction in continuous domains.
>
> Our Chain domain corresponds to the "tied" version of Chain in Poupart et al. When we match their setup to compare results (and report 95% confidence intervals rather than standard deviation), BPO performs similarly to the discrete BRL and POMDP algorithms.
> BPO, Chain-10: 3645.6 ± 5.4
> Beetle (discrete BRL): 3650 ± 3.6
> Perseus (discrete POMDP solver): 3661 ± 2.4
>
> We also add a new experiment for the more challenging "semi-tied" version, where the slip probabilities for the two actions are estimated independently. Again, BPO performs comparably to discrete BRL and POMDP algorithms.
> BPO, Chain-10: 3649.1 ± 7.8
> Beetle: 3648 ± 3.7
> Perseus: 3651 ± 2.8
> MC-BRL, K=10 (Wang et al., 2012; continuous BRL): 3216 ± 64
>
>
> Encoding did not improve performance significantly:
> We respectfully disagree with the assessment that "the encoding did not improve the performance significantly," which seems to conflict with the later observation that "the proposed algorithm, especially with encoders, is quite robust w.r.t. the discretization." The experiment comparing performance on different levels of discretization on the Chain environment is specifically designed to evaluate the usefulness of the encoding. Figure 2b demonstrates that the separate encoder networks improve the performance of BPO compared to BPO- when the latent space is finely discretized. We would welcome further clarifications to better understand this concern.
>
>
> Updating distribution over MDPs:
> The transition functions of the MDPs are parameterized by latent variables. The Bayes filter maintains a posterior distribution over the latent parameters, which is equivalent to a distribution over MDPs. In Section 5, we further describe how we implement a Bayes filter for the uniformly discretized latent parameter space and mention that other Bayes filters such as the extended Kalman filter can be used for Gaussian belief distributions.
>
>
> Improve structure of paper:
> Thank you for your helpful suggestions about which sections to pare down. We will incorporate them into the next draft.
>
>
> Poupart, Vlassis, Hoey, Regan. An Analytic Solution to Discrete Bayesian Reinforcement Learning. ICML 2006.
> Wang, Won, Hsu, Lee. Monte Carlo Bayesian Reinforcement Learning. ICML 2012.
> Platt, Tedrake, Kaelbling, Lozano-Perez. Belief space planning assuming maximum likelihood observations. RSS 2010.

---

### Author Response · Authors · 2018-11-25
**Revised paper with better visualization and additional technical details submitted**

We thank all reviewers for their thoughtful feedback and comments.

Our paper provides a scalable RL algorithm for addressing model uncertainty. Our algorithm is a solid combination of Bayes filter, Monte Carlo methods, and batch policy optimization algorithms [Reviewer 2], which we extend with a novel policy structure [Reviewer 3] to address the challenge of large latent state spaces. As pointed out by all three reviewers, our experiments demonstrate promising results for continuous BAMDP and POMDPs.

Our algorithm depends on a fixed-size parameterization of the continuous latent state space, e.g. a mixture of Gaussians. When such a specific representation is not appropriate, we can choose a more general uniform discretization of the latent space.

Discretizing the latent space introduces an additional challenge from the curse of dimensionality. It is crucial that an algorithm is robust to the size of the belief representation. Our belief encoder (Section 4) achieves the desired robustness by learning a compact representation of an arbitrarily large belief representation. We empirically verify that the belief encoder makes our algorithm more robust to large belief representations than the one without the belief encoder (Figure 2).

Here is a summary of our revisions:
An additional figure that compares the relative performance of BPO with other approaches (Figure 2a). The table of numerical results has been moved to the Appendix.
An additional figure that demonstrates the importance of the belief encoder (Figure 2b)
Reviewer 1: Visualization of entropy reduction and performance on the MuJoCo BAMDP problems (Figure 4)
Reviewer 1 & 2: Clarification of belief representation (Section 4)
Reviewer 1 & 2: A new section on Bayes filter (Section 5)
Reviewer 3: Experimental details about the policy network and training parameters (Section 7, Appendix).
Reviewer 1: Reference to DVRL added in Section 3.

---

### Public Comment · (anonymous) · 2018-12-14
**Important baselines missing**

If I understand correctly, the work considers a distribution over MDPs (i.e. belief state) and attempts to optimize policies over the belief state in a tractable way. The straightforward baseline seems to be the work of Peng et al. [1] which considers the same problem statement and trains an LSTM policy with PPO. The LSTM can be interpreted as implicitly performing inference and action selection simultaneously. Are the problem statements in the papers different? If not, how would the proposed BPO method compare against an LSTM policy?

Also, for the EPOpt baseline, is the belief distribution updated (on a per-time-step basis) and policy re-optimized? If not, it seems straightforward that a static algorithm would not perform as well as an adaptive algorithm, when the performances are measured either during the process of adaptation or after adaptation. Thus, a better comparison is EPOpt with a recurrent policy (similar to [1]) or EPOpt with feedforward policies but beliefs and policies are updated after getting data from the target domain.

[1] Sim-to-real transfer of robotic control with dynamics randomization, Peng et al. ICRA 2018.

---

> ### Author Response · Authors · 2018-12-17
> **Reply**
>
> Thank you for your feedback.
>
> BPO vs. Peng et al. [1]:
> The posterior belief distribution compactly summarizes the history of observations, and LSTMs can be interpreted similarly. The key difference between BPO and [1] is that BPO explicitly utilizes the belief distribution, while in [1] the LSTM must implicitly learn an embedding for the distribution. Because the belief distribution is directly provided to a BPO policy, a feed-forward policy is sufficient.
>
> BPO’s explicit use of a Bayes filter leads to data-efficiency and interpretability. As the learner does not need to infer the distribution from a long history of observations, the policy search can be more data-efficient. A Bayes filter also improves interpretability, as we can directly test how the agent adapts to different beliefs. However, such a Bayes filter may not be always readily available, e.g. when the latent parameters are only partially identifiable. We believe that combining a recurrent policy for unidentified parameters (analogous to [1]) with a Bayes filter for identified parameters (as in BPO) would be an interesting future direction to pursue.
>
> EPOpt baseline:
> The EPOpt baseline we provide is belief-agnostic, as we acknowledge in Section 7: “... we  emphasize that neither (EPOpt nor UP-MLE) formulate the problems as BAMDPs.” The main reason for using this baseline is to analyze when belief-awareness is truly necessary. As we demonstrate through the MuJoCo experiments, belief-agnostic learners (EPOpt, UP-MLE) can perform quite well when active information-gathering is not critical. The adaptive version of EPOpt proposed in their paper assumes **multiple episodes of interaction** with the target domain with which they update the source distribution and retrain the agent, which is very different from our problem setting in which only **one episode of interaction** is allowed at test time. Since BPO can also be interpreted as belief-aware domain randomization, and it is similar to EPOpt if a per-step belief update is used as its input.

---

### Meta-Review · Area_Chair1 · 2018-12-14
**Useful combination of existing techniques**

**Confidence:** 3
**Recommendation:** Accept (Poster)

**Metareview:**

The paper proposed a deep, Bayesian optimization approach to RL with model uncertainty (BAMDP).  The algorithm is a variant of policy gradient, which in each iteration uses a Bayes filter on sampled MDPs to update the posterior belief distribution of the parameters.  An extension is also made to POMDPs.

The work is a combination of existing techniques, and the algorithmic novelty is a bit low.  Initial reviews suggested the empirical study could be improved with better baselines, and the main idea of the proposed method could be expended.  The revised version moves towards this direction, and the author responses were helpful.  Overall, the paper is a useful contribution.